# Improved Pharmacokinetics of Icariin (ICA) within Formulation of PEG-PLLA/PDLA-PNIPAM Polymeric Micelles

**DOI:** 10.3390/pharmaceutics11020051

**Published:** 2019-01-25

**Authors:** Lu-Ying Han, Yun-Long Wu, Chun-Yan Zhu, Cai-Sheng Wu, Chun-Rong Yang

**Affiliations:** 1College of Pharmacy, Jiamusi University, 148 Xuefu Road, Jiamusi 154007, China; luyinghan1993@163.com; 2Fujian Provincial Key Laboratory of Innovative Drug Target Research, School of Pharmaceutical Sciences, Xiamen University, Xiamen 361005, China; wuyl@xmu.edu.cn (Y.-L.W.); 15671682663@163.com (C.-Y.Z.)

**Keywords:** polymeric micelles, icariin, bioavailability, extended-release

## Abstract

Icariin (ICA) is a major flavonoid that contains the active compound *Epimedii Folium*. However, ICA’s pharmacokinetic characteristics remain unsatisfactory due to its low bioavailability, and hence limited drugability. In order to improve its pharmacokinetics and achieve prolonged blood circulation time, a novel polymeric micelle, made of the self-assembled micelle between poly (ethylene glycol)-poly (_L_-lactic acid) (PEG-PLLA) and poly (_D_-lactic acid)-poly(*N*-isopropylacrylamide) (PDLA-PNIPAM), was designed to encapsulate ICA. Our experimental results showed that this polymeric micelle formulation of ICA exhibited uniform nano-size distribution and high stability within 48 h. The new formulation also allowed sustained ICA release in an in vitro drug release study. Furthermore, in vivo experiments revealed that ICA bioavailability in the PEG-PLLA/PDLA-PNIPAM polymeric micelle formulation was significantly higher compared to ICA alone, or ICA in the traditional Pluronic F127 micelle formulation. Finally, we show that metabolite analysis confirmed that ICA within the PEG-PLLA/PDLA-PNIPAM polymeric micelle formulation provided better drug protection, reduced drug metabolites production, and decreased undesired first-pass effects. Overall, these data show that ICA within PEG-PLLA/PDLA-PNIPAM polymeric micelle formulation exhibit advantages, in terms of improved physicochemical properties, sustained release of ICA in vitro, and improved bioavailability of ICA in vivo, which represent a feasible approach for improving the drugability of pharmaceutical small molecules with low bioavailability or poor stability.

## 1. Introduction

*Epimedii Folium* is the dry aerial part of various plants of the genus *Epimedium brevicornu* Maxim of Berberidaceae. In addition to its pungent nature and sweet flavor, it also possesses the functions of tonifying the kidneys [1], muscle or bone strengthening [2], dispelling rheumatism [3], etc. Clinically, it is commonly applied for the treatment of impotence, seminal emission, weakness of muscles and bones, rheumatic arthralgia, anti-tumor, numbness, and contracture [4,5,6,7,8,9,10]. Icariin (ICA) is the main active compound of *Epimedii Folium* [11]. Pharmacological studies have shown that ICA functions include anti-osteoporosis, enhancing immune function, protecting cardio-cerebrovascular systems, bacteriostasis, anti-inflammatory, anti-viral, anti-cancerous, and anti-aging effects [12,13,14,15,16,17,18]. However, further pharmaceutical development of ICA has been greatly hindered by its disadvantages, such as poor pharmacokinetics or low bioavailability.

Previous reports have shown that ICA absorption is limited when it enters the body in the form of a prototype drug. However, it can be absorbed into the body through the decomposition of bacterial enzymes in the gastrointestinal tract and intestinal flora into the metabolite icaritin, followed by phase II metabolism to produce glucuronic acid metabolites, which was the main reason for the poor absorption and low bioavailability of ICA in vivo [19]. Nanoparticle-based drug delivery system designs (i.e., in the form of ICA-propylene glycol (PG)-lipsome) have been shown to improve ICA pharmacokinetics, increase ICA concentrations in plasma and tissue, and prolong mean retention time (MRT), and biological half-life [20].

Polymeric micelles—a biocompatible nanoparticle delivery system—are thermodynamically stable, colloidal solutions formed by amphiphilic block copolymer self-assembly in water. Amphiphilic polymer micelles consist of hydrophilic and hydrophobic segments. The volume of hydrophilic segments is generally larger than that of hydrophobic segments, which can form core-shell structure in solution spontaneously. The micelle consists of a core and shell. The core is formed by hydrophobic segments (i.e., saturated fat chains, poly(lactic acid), and poly(caprolactone)), while the shell is formed by hydrophilic segments (i.e., chitosan, poly(ethylene glycol) or PEG, and hyaluronic acid), or temperature-sensitive material (poly(*N*-isopropyl acrylamide) or PNIPAM). This unique structure would allow loading the core with hydrophobic drugs, thereby reducing the drug’s toxicity or other side effects. The hydrophilic shell wraps the drug, and thus plays a protective role that can improve the drug’s stability and achieve its sustained release [21,22,23,24]. Therefore, in this study, we chose polymeric micelles as drug carriers to further improve the pharmacokinetic properties of ICA in vivo.

F127 is a biodegradable material approved by the FDA with wide applications in nano-medicine. ICA’s formulation was established by comparison to that of the traditional Pluronic F127 (block copolymer of poly(ethylene glycol)-poly(propylene glycol)-poly(ethylene glycol) or PEG-PPG-PEG) polymeric micelles. Therefore, novel poly(ethylene glycol)-poly(_L_-lactic acid)/poly(_D_-lactic acid)-poly(*N*-isopropylacrylamide) (PEG-PLLA/PDLA-PNIPAM) polymeric materials were integrated into the pharmaceutical formulation of ICA. It is noteworthy that, in PEG-PLLA/PDLA-PNIPAM polymeric material, PNIPAM is a temperature-sensitive segment with the lowest critical solution temperature (LCST), while the shell chain consisted of random-coil structures (hydrophilic state) below the critical solution temperature. Temperatures above LCST caused polymer dehydration and spherical structure collapse (hydrophobic state), which led to micelle aggregation and enhanced stability of this micelle’s formation. In this report, to investigate whether PEG-PLLA/PDLA-PNIPAM polymeric nanomicelle formulation could have significant improvements on the pharmacokinetic properties of ICA, we compared the physical, chemical, and in vivo pharmacokinetic properties of PEG-PLLA/PDLA-PNIPAM and traditional F127 micelles. 

## 2. Materials and Methods

### 2.1. Materials

Tin 2-ethylhexanoate [Sn(Oct)_2_ 95%], toluene (99.8%, anhydrous), methanol (99.8%, anhydrous), 2-bromoisobutyryl bromide (98%), 2-(dimethylamino)ethyl methacrylate (99%), 1,4-dioxane (99.8%, anhydrous), copper bromide (I) (CuBr 99%), 1,1,4,7,10,10-hexamethyltriethylenetetraamine (dHMTETA, 97%), alumina (neutral), *N*-isopropyl (NIPAAm, 97%), trisodium citrate dihydrate, sodium acetate (anhydrous, >99%), ethylene glycol (anhydrous, 99%), ethanol (>99 %), acetonitrile (anhydrous, 99.8%), and tetrahydrofuran (anhydrous, 99.8%) were purchased from Sigma-Aldrich (St. Louis, CA, USA). Benzyl alcohol and chloroform were distilled on calcium hydride (CaH_2_) before use. _D_-Lactide (_D_-LA) and _L_-lactide (_L_-LA) were purchased from Purac Biochem (Gorinchem, Netherlands). Epimedium (>98%) was purchased from Baoji Chenguang Biotechnology Co., Ltd. (Baoji, China); F127 was purchased from the St. Louis, CA, USA; Wistar rats were purchased from the Xiamen University Animal Center (Xiamen, China).

### 2.2. Synthesis of Copolymers.

#### 2.2.1. Synthesis of (poly (_L_-lactic acid)) (PLLA), (poly (_D_-lactic acid)) (PDLA) and (poly (_D_-lactic acid)-poly (*N*-isopropylacrylamide)) (PDLA-PNIPAAm) 

The synthesis of PLLA and PDLA was according to previous reports [25]. _L_-lactide or _D_-lactide monomer (7.5 g) were mixed with Sn(Oct)_2_ (75 μL) in the solvent of ethanol (0.12 mL) and toluene (40 mL), within N_2_ environment for ring-opening polymerization. The mixture was stirred for 24 h, before applying cold methanol precipitation for purification, and then vacuum drying. Further modification of PDLA with conjugation of PNIPAAm was conducted using PDLA-Br as an initiator. More specifically, the terminal -OH group of PDLA was modified by 2-bromoisobutyryl bromide in chloroform, while the purification was conducted by precipitation in methanol and vacuum drying. Furthermore, PDLA-Br (1 g) was mixed with HMTETA (100 μL) in 1,4-dioxane (18 mL), and trace amounts of CuBr in N_2_ atmosphere. The polymerization lasted for a day at 75 °C before terminating by adding THF. Purification was conducted using an alumina column, and precipitation in hexane or diethyl ether to render the resultant PDLA-PNIPAAm diblock copolymer.

#### 2.2.2. Synthesis of (poly (ethylene glycol)-poly (_L_-lactic acid)) (PEG-PLLA)

PEG (5.0 g) was mixed with PLLA (2.6 g) and Sn(Oct)_2_. The mixture was vacuum dried and heated at 120 °C for 2 h. The resulting copolymer was dissolved in CH2Cl2 and re-precipitated into excess ether. Finally, a PEG-PLLA diblock copolymer having a yield of 94% was obtained.

#### 2.2.3. Synthesis of PEG-PLLA/PDLA-PNIPAM

PEG-PLLA and PDLA-PNIPAM were separately dissolved in tetrahydrofuran and mixed. The obtained solution was stirred at 25 °C for 0.5 h, and the solution was evaporated at room temperature (Figure 1). 

### 2.3. Preparation of Icariin Mixed Polymer Micelles

PEG-PLLA/PDLA-PNIPAM polymer carrier (10 mg) and 1 mg icariin were accurately weighed, and dissolved in 1 mL tetrahydrofuran by ultrasonic dissolution. The solution was slowly added, dropwise to 10 mL of distilled water while stirring at room temperature. Tetrahydrofuran was then removed by volatilization. The drug-loaded mixed polymer micelles were prepared. The reaction process is shown in Figure 2.

### 2.4. Determination of Particle Size and Zeta Potential

Particle size distribution and zeta potential of the drug-loaded micelles were measured using a laser-assisted particle size analyzer. Micelles were diluted to the appropriate concentrations with ultrapure water, and added to the sample cell to determine particle size, size distribution, and zeta potential.

### 2.5. Determination of Encapsulation Efficiency and Drug Loading

Using high-speed centrifugation, 1 mL of the micellar solution was centrifuged at 15,000 r/min for 10 min. The supernatant was collected, and its concentration was determined using high performance liquid chromatography at 270 nm. Encapsulation efficiency and drug loading were calculated according to the following formula:$$EE\%=(1-\frac{C_{free}}{C_{total}})\times 100\%$$
$$DL\%=\frac{W_{drug}}{W_{lipid}}\times 100\%$$

### 2.6. Investigation of Polymer Micelle Stability

To evaluate the stability of polymer micelles, particle size distributions of PEG-PLLA/PDLA-PNIPAM and F127 polymer micelles at 0 h, 6 h, 12 h, and 48 h were measured with a laser particle size analyzer at pH 1.2, 6.8, and 7.4, respectively.

### 2.7. Drug Release In Vitro

The in vitro release behavior of ICA, F127-ICA, PEG-PLLA/PDLA-PNIPAM-ICA solutions was investigated by dialysis, and phosphate buffer saline (PBS) buffer solutions with pH 1.2, 6.8, or 7.4 were selected as the medium. F127-ICA and PEG-PLLA/PDLA-PNIPAM-ICA polymer micelles containing equal amounts of ICA were accurately added to the dialysis bag, and tightly placed in PBS medium. At each time point, 1 mL of each group of drugs was placed at −4 °C for testing, while 1 mL of isothermal PBS was added to the beaker. The drug content in the release medium was determined by high performance liquid phase, and the cumulative release was calculated according to the formula:$$E_{r}=\frac{V_{e}\sum_{1}^{n-1}C_{i}+V_{0}C_{n}}{m_{drug}}$$

### 2.8. Cell Viability

The cell viability MTT (3-(4,5-dimethyl-2-thiazolyl)-2,5-diphenyl-2-H-tetrazolium bromide) assay was used to test the cytotoxicity of blank and drug-loaded micelles. To evaluate viable cell numbers, samples were added at varying concentrations 20, 40, 60, 80, 100, 200, and 500 μg/mL (*n* = 6), while the enzyme standard instrument absorbance value was 490 nm.

### 2.9. Pharmacokinetics of Icariin PEG-PLLA/PDLA-PNIPAM Micelles

#### 2.9.1. Liquid Phase Conditions

Analysis was performed using an Agilent 6460 Triple Quadrupole Mass Spectrometer (Palo Alto, CA, USA). Column: Angilent Poroshell 120 EC-C18 column (2.1 × 50 mm, 2.7 μm); column temperature: 40 °C; flow rate 0.1 mL/min; injection volume 3 μL; mobile phase: 0.1% formic acid aqueous solution (solution A) and acetonitrile (B solution), gradient elution: 0–2.0 min, 5% B; 2.0–4.0 min, 20% B; 4.0–10.0 min, 35% B; 10.0–16.0 min, 50% B; 16.0–18.0 min, 80% B; 18.0–19.0 min, 95% B; 19.0–19.5 min, 95% B; 19.5–24.0 min, 5% B.

#### 2.9.2. Mass Spectrometry Conditions

Electrospray ion source: ESI source; scanning mode: ESI positive ion mode; dry gas pressure: 40 psi; dry gas flow rate: 11 L/min; spray voltage: 35 psi; dry gas temperature: 350 °C; capillary voltage: 4000 V; and monitoring mode: Multiple Reaction Monitoring Mode (MRM).

#### 2.9.3. Animal Experiments

The animal experiment part of this project has been reviewed by the ethical committee of experimental animal management of xiamen university. Nine male wistar adult rats (200–240 g) were used for the study. The animals were fasting for 12 h before dosing and free water drinking. Icariin (20 mg/kg) was administered intragastrically, to investigate its pharmacokinetic characteristics. Three groups; A, B, and C referred to ICA standard solution, F127-ICA, and PEG-PLLA/PDLA-PNIPAM-ICA, respectively. Before the experiment, blank blood was collected from the fundus veins at 0 min, 15 min, 45 min, 1.5 h, 3 h, 6 h, 9 h, 12 h, 18 h, 24 h, 30 h, and 36 h after intragastric administration. Plasma samples were stored at –80 °C for testing. The time–concentration curve was plotted, while pharmacokinetic parameters, including the area under the concentration–time curve (AUC) and maximum plasma concentration (*C*_max_), were estimated by means of a non-compartmental analysis using Drug and Statistics 3.0 (DAS 3.0).

## 3. Results

### 3.1. Determination of Particle Size and Zeta Potential

The particle size distribution and zeta potential of PEG-PLLA/PDLA-PNIPAM and F127 polymer micelles were measured using a Mastersize Nano-ZS90 laser particle size analyzer. As shown in Table 1, the average particle sizes of PEG-PLLA/PDLA-PNIPAM and F127 were 128.5 ± 4.9 nm and 177.8 ± 3.2 nm, while the average zeta potentials were −3.5 ± 0.6 mV and −3.3 ± 0.7 mV, respectively. PEG-PLLA/PDLA-PNIPAM showed lower PDI values compared to F127. The results indicated that PEG-PLLA/PDLA-PNIPAM particle size might be more suitable for cell penetration, and more beneficial for exerting drug efficacy. In addition, comparing PDI values suggested that PEG-PLLA/PDLA-PNIPAM micelles were more uniform than their F127 counterparts in terms of particle size distribution. These experimental characterizations suggest superiority of PEG-PLLA/PDLA-PNIPAM over the traditional F127 polymeric micelles both in particle size and uniform distribution.

### 3.2. Determination of Entrapment Efficiency and Drug Loading

PEG-PLLA/PDLA-PNIPAM and F127 drug-loaded polymer micelles were prepared as described in the Methods section. Encapsulation efficiency and drug loading were measured by high performance liquid chromatography (HPLC; Table 2). The encapsulation efficiency of PEG-PLLA/PDLA-PNIPAM and F127 were 85.8 ± 1.9% and 70.9 ± 2.2%, while drug loading was 7.7 ± 0.2% and 6.5 ± 1.8%, respectively. These data indicate that PEG-PLLA/PDLA-PNIPAM shows strong drug encapsulation ability, and slightly better encapsulation efficiency or drug loading compared to F127.

### 3.3. Stability of Polymeric Micelles

The stability of polymer micelles after 48 h was evaluated by using dynamic light scattering to quantify their particle size changes (Figure 3 and Figure 4). PEG-PLLA/PDLA-PNIPAM polymer micelle particle size changes in pH 1.2, 6.8, and 7.4 were uniform and stable within the 48 h time period. However, F127 polymer micelle particle size distribution at pH 7.4 was not sufficiently homogeneous compared to its PEG-PLLA/PDLA-PNIPAM counterparts, and changed significantly after 12 h. These data show that while PEG-PLLA/PDLA-PNIPAM polymer micelles’ particle size did not change significantly over time under different conditions, F127 polymer micelles’ particle size and distribution changed significantly. The above findings suggest that PEG-PLLA/PDLA-PNIPAM micelles were stable in structure and could not be destroyed easily, which might effectively protect the encapsulated drugs in vivo and improve drug stability.

### 3.4. In Vitro Release

In vitro releases of ICA monomer solution, ICA–F127 and ICA–PEG-PLLA/PDLA-PNIPAM polymeric micelles at pH 1.2, 6.8, or 7.4 were shown in Figure 5. The ICA monomer solution was completely released at 12 h, while both ICA–F127 or ICA–PEG-PLLA/PDLA-PNIPAM could play a role in sustained release. Furthermore, ICA–PEG-PLLA/PDLA-PNIPAM-ICA exhibited a more obvious sustained release effect and might protect the drug for a longer period compared to the ICA–F127 formulation. 

### 3.5. Cell Viability

The cytotoxicity of PEG, F127, ICA–F127, and ICA–PEG-PLLA/PDLA-PNIPAM micelles were investigated at indicated concentrations. The results are shown in Figure 6. The results showed that the toxicity of blank micelles was less than that of drug carrier micelles, and that the toxicity of PEG-PLLA/PDLA-PNIPAM micelles was less than that of F127 under the same conditions regardless of drug carrier.

### 3.6. Pharmacokinetics of ICA Loaded Polymeric Micelles

The pharmacokinetics of ICA-loaded polymeric micelles were performed using quantitative analysis of its blood concentration. The peak area of blood samples was measured using a triple quadrupole mass spectrometer, and the concentration was calculated according to the standard curve. The time–drug curve is presented in Figure 7. Plasma concentrations of ICA in the PEG-PLLA/PDLA-PNIPAM polymer micelle formulation decreased slowly, and were significantly higher than those of the other two groups. Drug parameters’ calculations were obtained using Das3.0 software. PEG-PLLA/PDLA-PNIPAM, as a polymer carrier, effectively increased the exposure of prototype drugs in vivo and prolonged the drugs’ therapeutic effects (Table 3). The relative bioavailability of PEG-PLLA/PDLA-PNIPAM was 216% of F127, and 500% of the ICA monomer. Additionally, the main metabolites’ analysis results (Appendix A) showed that the amount of ICA metabolites encapsulated by PEG-PLLA/PDLA-PNIPAM polymer micelles was remarkably lower than that of the other two groups, and the ICA encapsulated by PEG-PLLA/PDLA-PNIPAM polymer micelles was more stable in vivo (metabolite concentrations were aligned using the ICA standard curve).

## 4. Discussion

After oral administration, Icariin (ICA) is easily hydrolyzed by bacteria in the intestinal tract into baohuoside I and icaritin, thus significantly reducing ICA’s bioavailability. Nano-polymeric micelles can improve the permeability of drugs through membranes, and effectively avoid drug loading digestion by intestinal bacteria, which might serve as an important approach for improving drug bioavailability. In this experiment, PNIPAM in the synthesized PEG-PLLA/PDLA-PNIPAM polymer carrier was a temperature-sensitive polymer with LCST at 32 °C. Therefore, PNIPAM might change from hydrophilic state to hydrophobic state in vivo, and from outward extension to inward convergence (Figure 8). The physical status of PNIPAM segments could change with high temperature sensitivity, thus the PEG-PLLA/PDLA-PNIPAM polymer carrier could ensure that ICA is not hydrolyzed by intestinal flora. This process would ensure greater exposure in vivo compared to traditional material (i.e., Pluronic F127), and increase the total amount of ICA absorbed into the body. Furthermore, our study showed that applying the PEG-PLLA/PDLA-PNIPAM polymeric carrier ensures that ICA had a few metabolic transformations during the enterohepatic circulation (released from bile discharge into duodenum and thus reabsorbed into the body circulation). The blood drug concentration of the second absorption peak (*T*_max_ at 8 h) of this novel PEG-PLLA/PDLA-PNIPAM polymeric micelle formulation was significantly higher than that of the other two groups (in case of F127 as traditional polymeric micelle, or ICA monomer only; Figure 7), indicating the improved physicochemical or pharmaceutical properties of ICA within the PEG-PLLA/PDLA-PNIPAM polymeric micelle formulation. Consequently, the excellent properties of this novel drug delivery system with special functions provided more feasible methods of improvement for drugs with low bioavailability or poor stability.

## 5. Conclusions

In this study, PEG-PLLA/PDLA-PNIPAM was designed to encapsulate ICA in the formulation of polymeric micelles using the solvent evaporation method. Experimental results showed that PEG-PLLA/PDLA-PNIPAM micelle structure was more stable than that of the traditional Pluronic F127, which could better protect the encapsulated ICA and increase its absorption in vivo, thus ameliorating the issue of rapid metabolism of ICA in vivo and improving its bioavailability.

## Figures and Tables

**Figure 1 pharmaceutics-11-00051-f001:**
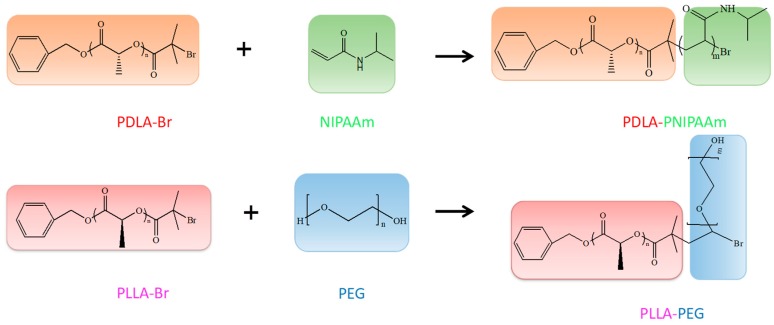
(Poly(ethylene glycol)-poly(_L_-lactic acid))/(poly(_D_-lactic acid)-poly(*N*-isopropylacrylamide)) (PEG-PLLA/PDLA-PNIPAM) synthetic roadmap.

**Figure 2 pharmaceutics-11-00051-f002:**
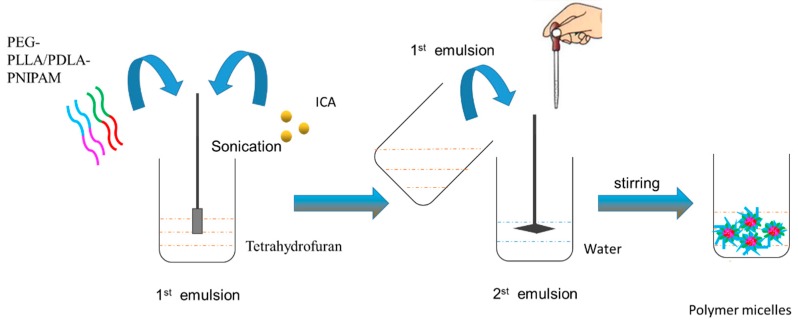
Preparation of drug-loaded micelles.

**Figure 3 pharmaceutics-11-00051-f003:**
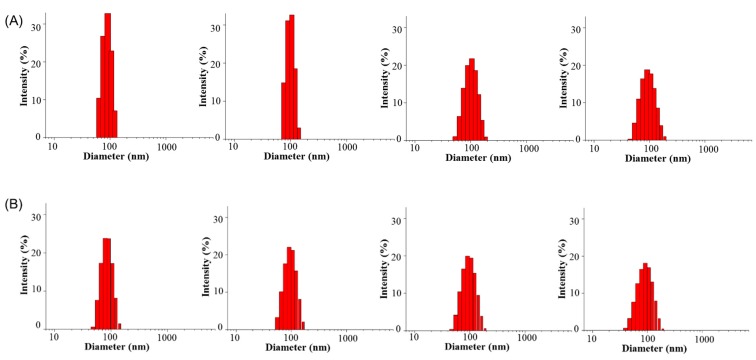
Particle size distribution of PEG-PLLA/PDLA-PNIPAM polymer micelles in PBS solutions of pH 1.2, 6.8, and 7.2, respectively, in 48 h (from left to right are 0 h, 6 h, 12 h, and 48 h, respectively).

**Figure 4 pharmaceutics-11-00051-f004:**
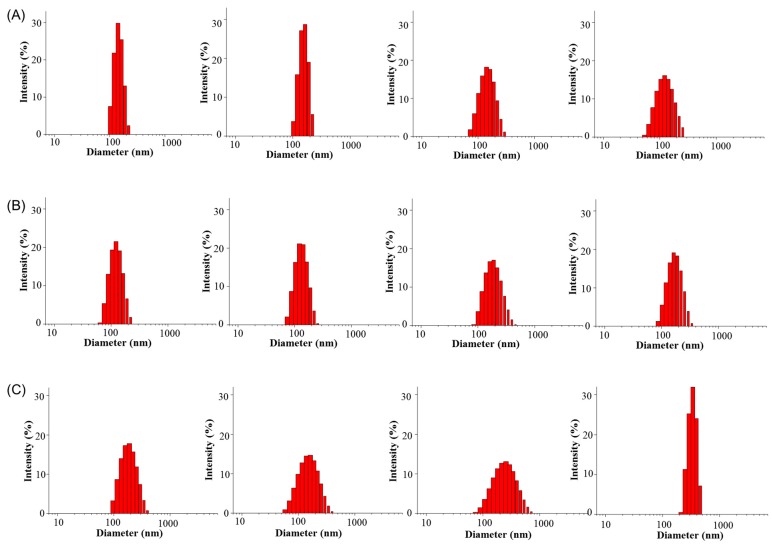
Particle size distribution of Pluronic F127 polymer micelles in 48 h (from left to right are 0 h, 6 h, 12 h, and 48 h, respectively).

**Figure 5 pharmaceutics-11-00051-f005:**
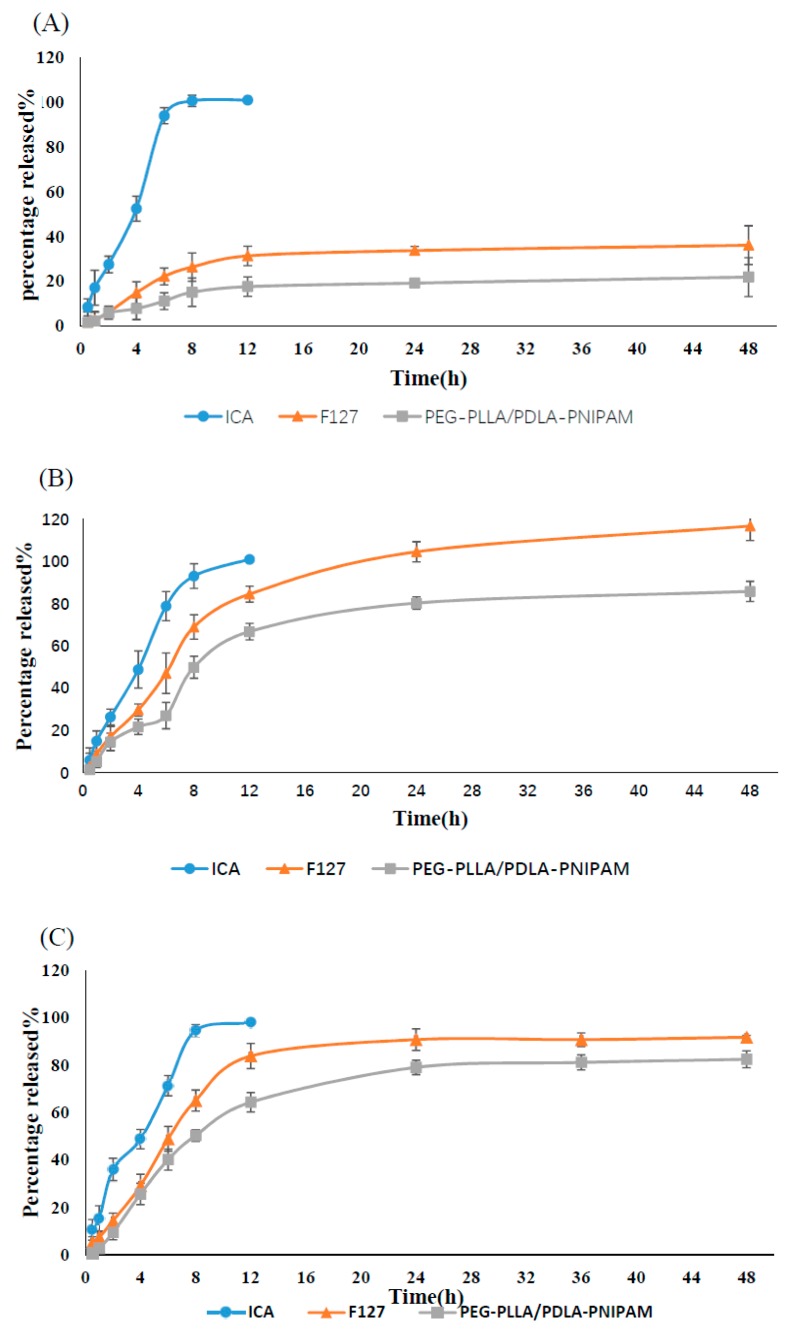
In vitro release profiles of ICA, ICA–F127, and ICA–PEG-PLLA/PDLA-PNIPAM in pH 1.2 (**A**), 6.8 (**B**), and 7.4 (**C**) buffer solutions, respectively (*n* = 3).

**Figure 6 pharmaceutics-11-00051-f006:**
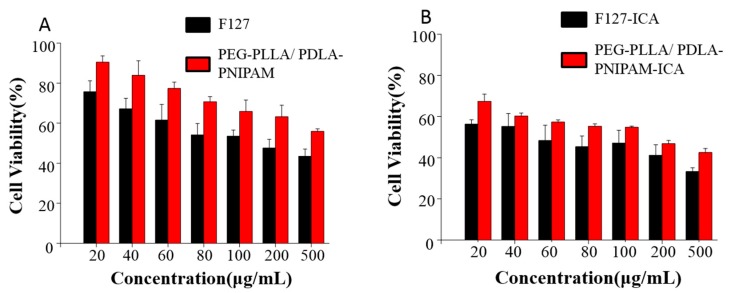
Cell viability of blank micelles (**A**) and drug-loaded micelles (**B**).

**Figure 7 pharmaceutics-11-00051-f007:**
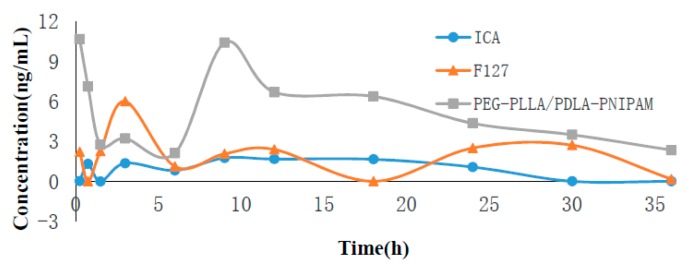
Icariin (ICA) concentration versus time after oral of different drug formulations (*n* = 3).

**Figure 8 pharmaceutics-11-00051-f008:**
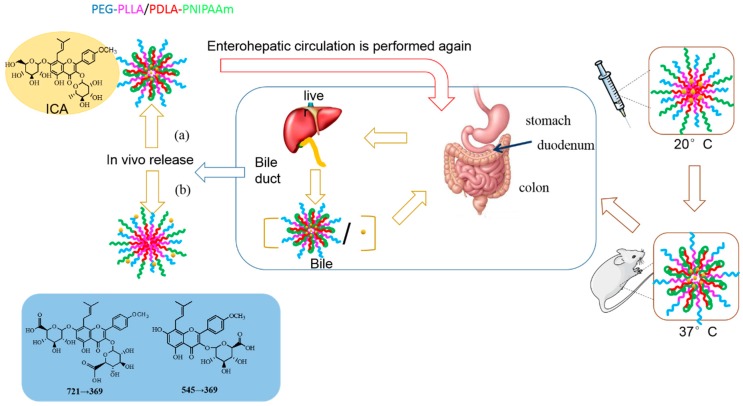
Morphological changes of PEG-PLLA/PDLA-PNIPAM in vivo.

**Table 1 pharmaceutics-11-00051-t001:** Polymer micelle particle size and potential measurement (*n* = 3).

Polymer Micelles	PDI	Particle Size (nm)	Zeta Electric
F127	0.28 ± 0.01	177.76 ± 3.23	−3.26 ± 0.67
PEG-PLLA/PDLA-PNIPAM	0.196 ± 0.02	128.53 ± 4.89	−3.50 ± 0.57

**Table 2 pharmaceutics-11-00051-t002:** Encapsulation efficiency and drug loading of PEG-PLLA/PDLA-PNIPAM drug-loaded micelles (*n* = 3).

Polymer Micelles	Encapsulation Efficiency (%)	Drug Loading (%)
F127	70.86 ± 2.19	6.45 ± 1.78
PEG-PLLA/PDLA-PNIPAM	85.76 ± 1.90	7.74 ± 0.17

**Table 3 pharmaceutics-11-00051-t003:** Pharmacokinetic parameters.

Polymeric Micelle	AUC (0–*t*) (μg/L * h) 676.9→368.9	*C*_max_ (μg/L) 676.9→368.9	AUC (0–*t*) (μg/L * h) 720.9→368.9	*C*_max_ (μg/L) 720.9→368.9	AUC (0–*t*) (μg/L * h) 544.9→368.9	*C*_max_ (μg/L) 544.9→368.9
ICA	35.78	1.76	235.18	25.09	20.90	1.83
F127-ICA	69.26	6.01	271.60	22.71	18.81	1.14
PEG-PLLA/PDLA-PNIPAM-ICA	179.03	10.67	133.75	6.29	17.45	1.09

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
