# Peer review of "Improved Pharmacokinetics of Icariin (ICA) within Formulation of PEG-PLLA/PDLA-PNIPAM Polymeric Micelles"

_pharmaceutics, 2019, doi:10.3390/pharmaceutics11020051_

Reviewer 1 Report

1)   In the 2nd paragraph, it has been mentioned that “…. it can be absorbed into the body through the decomposition of enzymes in the gastrointestinal tract and ….”. This paragraph needs to be much more specific. What enzymes? Which part of the GI tract? What is the target? Where is the main absorption site for this drug? etc.

2)   Section 2.4 is missing important details. What was the solution/medium used for measuring the zeta potential of the missiles? What was the pH? How can you make sure there will not be coagulation of the carriers in different environmental pH values?

3)   As the simulated intestinal fluid, PBS with pH 7.4 has been used, which is high compared to human intestinal pH values. In vivo results may not necessarily guarantee that you will have 100% release of the loaded drug. Also, how could you make sure about the end point of the release test? How could you make sure there is no remaining drug inside the non-releasing micelles? The data shown in figure 5 is just showing the plateau of release, which may not be the end point necessarily!

4)   What is the optimum size of the micelles for this application? What is the main absorption/uptake behavior considered for this delivery system?

5)   The loading capacity and encapsulation efficiency of the carriers have been measured in the absence of any environmental mechanical agitation. Also, authors have not reported how much activity loss may occur during the fabrication of the micelles?! How about possible gastric denaturation of the drug? Also, it would be meaningful to review the possible reasons for the huge gap between EE% and DL% of your system.

6)   What was the media used for evaluating the stability of the micelles? Was that comparable to gastric/intestinal/physiological environment? Why F127 is showing swelling or instability and your micelles are not?

Author Response

Dear reviewers,

Enclosed please find our recent revision entitled “Improved Pharmacokinetics of Icariin (ICA) within Formulation of PEG-PLLA/PDLA-PNIPAM Polymeric Micelle” which was intended to re-submit to pharmaceutics. The responses (red) to the comments made by the reviewers are included at the bottom of this letter.

We appreciate your time and effect in reviewing our manuscript. We have further conducted in vitro release and stability of the drug at different pH values, and cytotoxicity at different concentrations. experiments and re-organized the writing according to reviewers’ valuable comments. We hope that you will find this version suitable for publication. We are looking forward to hearing from you soon.

Yours Sincerely,                                

Caisheng Wu

Fujian Provincial Key Laboratory of Innovative Drug Target Research, School of Pharmaceutical Sciences, Xiamen University, Xiamen, China

Xiang’an South Road, Xiamen, China

Tel: +86-592-2187221   Fax: +86-592-2181879   E-mail: [email protected]

Chunrong Yang

College of Pharmacy, Jiamusi University, Heilongjiang, China

148 Xuefu Road, Jiamusi 154007, Heilongjiang, China

Tel: +86-15245451696       E-mail: [email protected]

Response to Reviewer 1 Comments

Point 1: In the 2nd paragraph, it has been mentioned that “…. it can be absorbed into the body through the decomposition of enzymes in the gastrointestinal tract and ….”. This paragraph needs to be much more specific. What enzymes? Which part of the GI tract? What is the target? Where is the main absorption site for this drug? etc.

Response 1: We thank the reviewer for your suggestion, icariin can be absorbed into the body through the decomposition of bacterial enzymes in the gastrointestinal tract. It is difficult to exist in the body with prototype drugs, and it is easy to undergo metabolic conversion in the gastrointestinal tract, which is the main reason for the low bioavailability of icariin. The content is supplemented in lines 43-45.

Point 2: Section 2.4 is missing important details. What was the solution/medium used for measuring the zeta potential of the missiles? What was the pH? How can you make sure there will not be coagulation of the carriers in different environmental pH values?

Response 2: We agree with the reviewer and the particle size change of the polymer micelles at different pH values was supplemented in section 2.6.

Point 3: As the simulated intestinal fluid, PBS with pH 7.4 has been used, which is high compared to human intestinal pH values. In vivo results may not necessarily guarantee that you will have 100% release of the loaded drug. Also, how could you make sure about the end point of the release test? How could you make sure there is no remaining drug inside the non-releasing micelles? The data shown in figure 5 is just showing the plateau of release, which may not be the end point necessarily!

Response 3: In the experimental part of 2.7, we supplemented the drug release at different pH values, pH 1.2 and 6.8, respectively. The experimental results are shown in Figure 5.

Point 4: What is the optimum size of the micelles for this application? What is the main absorption/uptake behavior considered for this delivery system?

Response 4: Polymer micelles generally have an optimum size between 10 nm and 200 nm. The molecular weight of the polymer, stability, and preparation are key factors in determining the size of the particle. The polymer micelle protects the drug from being decomposed by the gastrointestinal enzymes, performs multiple liver and intestinal cycles and slowly releases the drug, thereby increasing the absorption of the drug in the body.

Point 5: The loading capacity and encapsulation efficiency of the carriers have been measured in the absence of any environmental mechanical agitation. Also, authors have not reported how much activity loss may occur during the fabrication of the micelles?! How about possible gastric denaturation of the drug? Also, it would be meaningful to review the possible reasons for the huge gap between EE% and DL% of your system.

Response 5: Thank you for point out this issue. The activity test was not the main focus of this manuscript. But we have evaluated the structure stability or changes in micelle formulation. The drug will remain in the stomach for about 2-3 hours. Figure S1 shows that the metabolite produced by the drug-loaded polymer micelles in the first three hours is much lower than that of ICA, indicating that the drug-loaded polymer micelles are largely metabolized and not denatured in the gastric. But we still thank you for your advice, and hope to report drug activity in different formulations by creating activity evaluation animal model in near future.

EE of ICA in PEG-PLLA/PDLA-PNIPAM polymer micelle is about 85%. In this experiment, drug carrier micelles were prepared with a ratio of 1:10, so DL%= (0.85mg/11mg) * 100%=7.7%.

Point 6:

What was the media used for evaluating the stability of the micelles? Was that comparable to gastric/intestinal/physiological environment? Why F127 is showing swelling or instability and your micelles are not?

Response 6: Section 2.6 supplements the stability test of pH 1.2 and 6.8, as shown in Figure 3-4.

In this experiment, PNIPAM in the synthesized PEG-PLLA/PDLA-PNIPAM polymer carrier was a temperature-sensitive polymer with LCST, and its LCST temperature was 32 ° C. Therefore, PNIPAM might change from hydrophilic state to hydrophobic state in vivo, and PNIPAM might change from outward to inward to make the structure more stable, as shown in Figure 8.

Reviewer 2 Report

line-34-35: 

the authors list a series of effects, but only one reference is reported for only one of these effects. the sentence should be reviewed or more references should be added for the effects mentioned.

line 37: 

the authors should find a more specific reference for the clinical use of "Epimedii Folium"

for the in vivo study the authors should specify, 

how was the dose chosen? 

specify the dose for each experimental group?

what was the vehicle for the administration for the three different groups?

the data shown (n = 3) is the average of 3 tests or the result of 3 sets of different experiments?

furthermore, in the materials and methods section, the authors could specify what kind of statistical analysis was performed ?

Have any toxicity studies on this new polymer been conducted?

Author Response

Dear reviewers,

Enclosed please find our recent revision entitled “Improved Pharmacokinetics of Icariin (ICA) within Formulation of PEG-PLLA/PDLA-PNIPAM Polymeric Micelle” which was intended to re-submit to pharmaceutics. The responses (red) to the comments made by the reviewers are included at the bottom of this letter.

We appreciate your time and effect in reviewing our manuscript. We have further conducted in vitro release and stability of the drug at different pH values, and cytotoxicity at different concentrations. experiments and re-organized the writing according to reviewers’ valuable comments. We hope that you will find this version suitable for publication. We are looking forward to hearing from you soon.

Yours Sincerely,                                

Caisheng Wu

Fujian Provincial Key Laboratory of Innovative Drug Target Research, School of Pharmaceutical Sciences, Xiamen University, Xiamen, China

Xiang’an South Road, Xiamen, China

Tel: +86-592-2187221   Fax: +86-592-2181879   E-mail: [email protected]

Chunrong Yang

College of Pharmacy, Jiamusi University, Heilongjiang, China

148 Xuefu Road, Jiamusi 154007, Heilongjiang, China

Tel: +86-15245451696       E-mail: [email protected]

Response to Reviewer 2 Comments

Point 1: line-34-35: the authors list a series of effects, but only one reference is reported for only one of these effects. the sentence should be reviewed or more references should be added for the effects mentioned.

Response 1: We thank the reviewer for your suggestion and more references are added in line 35.

Point 2: line 37: the authors should find a more specific reference for the clinical use of "Epimedii Folium"

Response 2: We agree with the reviewer and more references are added in line 37 for the clinical use of "Epimedii Folium".

Point 3: for the in vivo study the authors should specify, how was the dose chosen? specify the dose for each experimental group? what was the vehicle for the administration for the three different groups?

Response 3: The dosages of all three groups were 20mg/kg and the solvent of the drug were 0.9% normal saline. The choice of drug dosage is determined by the drug loading of the micelle, clinical dosage, and then pre-experiment is performed to verify the feasibility. The dosage was supplemented in section 2.9.3 of the article.

Point 4:

the data shown (n = 3) is the average of 3 tests or the result of 3 sets of different experiments?

furthermore, in the materials and methods section, the authors could specify what kind of statistical analysis was performed ?

Response 4: "n=3" for each experimental group, three rats were used as parallel experiments, and the plasma of each group was determined in combination. In addition, the statistical analysis method is added in the 173-176 lines of the article

Point 5: Have any toxicity studies on this new polymer been conducted?

Response 5: We agree with the reviewer and we have supplemented the cytotoxicity experiment, as shown in figure 6.

Round  2

Reviewer 1 Report

I think it is good enough to publish the current revision in Pharmaceutics.